# Cloning and Expression of a *Perilla frutescens* Cytochrome P450 Enzyme Catalyzing the Hydroxylation of Phenylpropenes

**DOI:** 10.3390/plants9050577

**Published:** 2020-05-01

**Authors:** Mariko Baba, Ken-ichi Yamada, Michiho Ito

**Affiliations:** 1Department of Pharmacognosy, Graduate School of Pharmaceutical Sciences, Kyoto University, Sakyo, Kyoto 606-8501, Japan; baba.mariko.26n@st.kyoto-u.ac.jp; 2Department of Pharmaceutical Organic Chemistry, Graduate School of Pharmaceutical Sciences, Tokushima University, Shomachi, Tokushima 770-8505, Japan; yamak@tokushima-u.ac.jp

**Keywords:** phenylpropanoid, volatile compound, molecular cloning, cytochrome P450, biosynthetic pathway, perilla

## Abstract

Phenylpropanoid volatile components in plants are useful and valuable not only as flavorings, but also as medicines and food supplements. The pharmacological actions and toxicities of these compounds have been well studied but their synthetic pathways are generally unclear. In this study, we mined expressed sequence tag libraries of pure strains of perilla maintained for over 30 years for their oil type and conducted gas chromatography-mass spectrometry analyses of the perilla oils to confirm the presence of monohydrates speculated to be intermediates of the phenylpropene synthetics pathways. These putative monohydrate intermediates and their regioisomers were synthesized to identify the reaction products of assays of heterologously expressed enzymes. An enzyme involved in the synthesis of a phenylpropanoid volatile component was identified in perilla. Expression of this enzyme in *Saccharomyces cerevisiae* showed that it is a member of the cytochrome P450 family and catalyzes the introduction of a hydroxy group onto myristicin to form an intermediate of dillapiole. The enzyme had high sequence similarity to a CYP71D family enzyme, high regiospecificity, and low substrate specificity. This study may aid the elucidation of generally unexploited biosynthetic pathways of phenylpropanoid volatile components.

## 1. Introduction

Perilla (*Perilla frutescens* Britton var. *crispa* W. Deane) is a common culinary herb in East and Southeast Asia, and types of perilla that principally contain perillaldehyde, a monoterpene (MT) compound, in their essential oils are used pharmaceutically in China and Japan. Essential oils of perilla can be classified into more than 10 types [1], which are roughly divided into two groups based on the structures of their constituents [2]: MT-type oils, and phenylpropene (PP)-type oils whose main constituents are elemicin, myristicin, dillapiole, and nothoapiole (Figure 1). This biosynthetic pathway of perilla oil is genetically controlled [3,4], and the functions of each gene can be determined by cloning the enzymes that catalyze the relevant reaction steps in the biosynthetic pathway. The enzymes that catalyze the formation of MT-type oils, such as limonene synthase, geraniol synthase, linalool synthase, and perillaldehyde synthase, were previously characterized [5,6,7], whereas no PP-type synthases have been reported to date. Known phenylpropanoid volatile components include anethole in fennel, apiole in parsley, eugenol in clove, and myristicin in nutmeg (Figure 1 and Figure 2), and their pharmacological actions and toxicities are well studied due to their use as pharmaceuticals and as flavorings. For example, myristicin is used to treat rheumatism and anxiety in traditional medicine. Risk assessment of myristicin using the margin of exposure approach has been conducted because of its potential genotoxicity or carcinogenicity [8]. Dillapiole, which is found mainly in dill, has been reported to have cytotoxic effects [9]. However, only a few enzymes involved in the synthesis of these compounds have been isolated and reported, and these known enzymes catalyze the formation of compounds with simple side groups [10].

Perilla plants, whose major aromatic compound is perillaldehyde, are used as herbal medicine for Kampo prescriptions in Japan. Those perilla plants that are rich in perillaldehyde do not contain (*E*)-asarone (Figure 2) [11]. (*E*)-Asarone is structurally similar to PP-type oil components [12,13] and is likely formed through a PP-type oil component pathway, although the details are currently unknown. However, the (*Z*)-asarone isomer was reported to be toxic [14] and (*E*)-asarone may be carcinogenic [15], and therefore it should always be certified that no perilla plants containing (*E*)-asarone are present in the plant material used to prepare the pharmaceutical formulation. In order to find plants including this toxic compound and remove them from medicine ingredients, purity tests using liquid chromatography, optimized to separate perillaldehyde and (*E*)-asarone, are defined in the Japanese Pharmacopoeia.

The main constituents of PP-type perilla oil are elemicin, myristicin, dillapiole, and nothoapiole, and are likely synthesized from phenylalanine (Figure 1). The reaction steps to all four compounds are genetically controlled based on crossing experiments using pure strains developed by repeated self-pollination, and on detailed gas chromatography (GC) analyses of perilla essential oils [2,16]. Previous reports on the synthesis of alkaloids suggested that methoxy or methylenedioxy groups are formed in the synthetic pathways leading to PP-type oil components and may involve hydroxylation or cyclization reactions by cytochrome P450 (hereafter, “P450”) [17]. P450 enzymes catalyze different reactions, such as oxidation, hydroxylation, epoxidation, and dealkylation, and are involved in the synthesis of many plant secondary metabolites. Their low substrate specificity and high reaction regio- and stereo-specificity are beneficial for generating useful compounds, making their catalytic mechanisms an attractive target of study.

Here we describe the isolation of a P450 enzyme involved in the synthesis of dillapiole in PP-type perilla oils using expressed sequence tag (EST) libraries of pure strains of perilla. The perilla strains have been developed and maintained for more than 30 years through repeated self-pollination. The high purity of these strains allows for the selection of P450 sequences that may be involved in the biosynthesis of each oil component by comparing the expression levels of those sequences [18].

## 2. Results

### 2.1. Isolation of a P450 Sequence from Perilla

P450-like sequences that are specifically expressed in perilla strains whose essential oils contain mainly dillapiole were identified. Contig 49487 was selected from our EST library because it had the largest difference in reads per kilobase of exon per million mapped read (RPKM) values of the sequences studied (Table 1). We PCR-amplified the selected contig using primers based on sequence information from the EST database, using complementary DNA (cDNA) as a template, then confirmed the size of the PCR product by agarose-gel electrophoresis. The size of contig 49487 was in accordance with the sequence information. The full-length sequence of contig 49487 was determined by RACE methods. The sequence *Pf-49487* (GenBank Accession No. LC476554) encoded 505 amino acids (57 kD, Figure 3), and regions characteristic of P450s, such as a proline-rich region, an oxygen-binding pocket, and a heme-binding region, were conserved [19]. BLAST analysis showed that the amino acid sequence of *Pf-49487* shared 56% identity with that of the *Sesamum indicum* premnaspirodiene oxygenase-like enzyme (predicted) (GenBank Accession No. JP653840) and this was the highest similarity among all entries in the database. Premnaspirodiene oxygenase is a hydroxylase classified as a member of the CYP71D family. *Pf-49487* had a 49% amino acid sequence identity to the characterized *Hyoscyamus muticus* premnaspirodiene oxygenase (GenBank Accession No. EF569601) [20]. Cinnamate 4-hydroxylase from *Arabidopsis thaliana* (GenBank Accession No. NP180607) was a member of the P450 family that hydroxylates the aromatic ring of phenylpropanoid compounds and shared 28% sequence identity with *Pf-49487* at the amino acid level. Based on the standardized P450 nomenclature system, *Pf-49487*, is in the CYP71D subfamily and was named CYP71D558.

### 2.2. Comparison of Pf-49487 Expression among Strains of Phenylpropene (PP)-Type Perilla

Quantitative reverse transcription-PCR (RT-PCR) was used to compare the *Pf-49487* expression levels in various strains using RNAs isolated from different oil types of perilla: strain 12 of PP-m, strain 25 of PP-emd, and strain 5316 of PP-mdn (Figure 1 and Figure 4). The difference in expression level among oil types is similar to the RPKM values in the EST library used for the isolation of *Pf-49487*, namely, strain 12: 0.0968, strain 25: 23.7, strain 5316: 56.7 (Table 1). These ratios are believed to be relevant to the dillapiole content (%) in oils, namely, strain 12: 0%, strain 25: 9.01%, strain 5316: 21.13% [4,16].

### 2.3. Heterologous Expression and Functional Analysis of Pf-49487

P450 enzymes with a bound carbon monoxide (CO) molecule show an absorption spectrum peak at ca. 450 nm, and heterologously expressed P450 enzymes with such an absorption maximum generally have high activity. The reduced CO-difference spectrum of the microsomal fraction of yeast harboring the *Pf-49487* expression plasmid was measured and a peak at 448 nm was observed, suggesting that *Pf-49487* has P450 activity (Figure 5).

The catalytic reaction of *Pf-49487* was analyzed using the microsomal fraction and myristicin as a substrate by GC-mass spectrometry (MS). The reaction product was 6-allyl-4-methoxy-1,3-benzodioxol-5-ol (Table 2, compound 20), a hydroxylated myristicin (Figure 6). When methyl eugenol or elemicin were used as substrates, the hydroxylated products 6-allyl-2,3-dimethoxyphenol (Table 2, compound 5) and 6-allyl-2,3,4-trimethoxyphenol (Table 2, compound 31) were obtained, respectively (Figure 6). Dillapiole, eugenol, limonene, and geraniol were not substrates for this enzyme (Table 3).

Although *Pf-49487* converted methyl eugenol to compound 5 (Table 3), compound 5 was absent in oil prepared from strain 5316, which expresses a higher level of *Pf-49487*, whereas 5-allyl-2,3-dimethoxyphenol (Table 2, compound 14), compound 20, 5-allyl-6,7-dimethoxy-1,3-benzodioxol-4-ol (Table 2, compound 30), and compound 31 were present. These results suggest that *Pf-49487* does not convert methyl eugenol to compound 5 in perilla.

The use of elemicin as a substrate resulted in lower activity than did the use of myristicin (Table 3). This result suggests that myristicin and not elemicin is the main substrate for *Pf-49487*.

We performed kinetic analyses of the *Pf-49487* reaction using myristicin as substrate. The P450 concentration in the microsomal fraction was determined to be 1.03 µM from the reduced CO-difference spectrum and the Km, turnover number (kcat), and catalytic efficiency (kcat/Km) values were 9.1 µM, 15 min^−1^, and 1.65 × 10^−6^ min^−1^ M^−1^, respectively. The Km value for the oxidization of premnaspirodiene by premnaspirodiene oxygenase (whose amino acid sequence is most similar to *Pf-49487*) was 14 µM, similar to that of *Pf-49487* [20]. The optimum pH of *Pf-49487* was approximately 7.5 (Figure 7), similar to that of premnaspirodiene oxygenase [20].

## 3. Discussion

*Pf-49487* is a P450 enzyme in perilla that introduces a hydroxy group onto myristicin to form a dillapiole intermediate (compound 20). *Pf-49487* catalyzes the hydroxylation of methyl eugenol and elemicin at a position similar to that of compound 20 but is unreactive towards limonene and geraniol, which are MT-type oil components, as well as dillapiole and eugenol, which are phenylpropanoid compounds. These results indicate that *Pf-49487* exhibits rather low substrate specificity and high regiospecificity. Premnaspirodiene oxygenase, whose sequence is most similar to that of *Pf-49487*, has similar properties [20]. Other members of the CYP71D family also have high regiospecificity, namely the limonene hydroxylases CYP71D13 and CYP71D18, which hydroxylate the C3 and C6 positions, respectively [21].

The biosynthetic pathways of PP-type oil components have been inferred from the results of crossing experiments conducted with pure strains of perilla containing elemicin, myristicin, dillapiole, and nothoapiole as the main oil components, and the genetic control of each step in the formation of elemicin, dillapiole, and nothoapiole was in accordance with Mendelism [16] (Figure 1). Phenylpropanoids are believed to be synthesized via the shikimate pathway and the genes for methyl eugenol formation have been cloned [22]; consequently, methyl eugenol could be a precursor of PP-type oil components. However, neither methyl eugenol nor compound 5 have been detected in perilla oil whereas *Pf-49487* produces compound 5 from methyl eugenol in vitro. On the other hand, compound 14 (a putative precursor of elemicin and myristicin) and compound 30 (a putative precursor of nothoapiole) were present in oil from strain 5316 (PP-mdn). However, compound 14 was not found in oil from strain 10 (PP-em) and strain 12 (PP-m), even though compound 14 is a likely intermediate of myristicin and thus was expected to be present in these oils. These findings indicate that the catalytic patterns and reactivities of enzymes may differ among oil types. Our identification of monohydrate intermediates of phenylpropanoid compounds in perilla oil suggests that *O*-methyltransferases or P450s are involved in perilla biosynthetic pathways and catalyze the generation of methoxy groups or methylenedioxy bridges from these monohydrate compounds. Our future studies will further explore these enzymes.

## 4. Materials and Methods

### 4.1. Plant Materials

All perilla plants used in this study were grown at the Experimental Station for Medicinal Plant Research, Graduate School of Pharmaceutical Sciences, Kyoto University. They have been bred and kept as pure lines through repeated self-pollination [18]. Strain numbers and oil types (Figure 1) of perilla used in this study were as follows: strain 10, PP-em; strain 12, PP-m; strain 16, PP-md; strain 25, PP-emd; strain 5316, PP-mdn; strain 5717, C.

### 4.2. Construction of the Expressed Sequence Tag (EST) Library and Cloning of Pf-49487

The EST library was constructed by the Kazusa DNA Research Institute as described previously [7]. P450-like sequences were selected from the EST library by BLAST analysis and the expression levels of these sequences were compared between different oil types. High expression level contigs in strains containing high levels of dillapiole (strains 16, 25, and 5316) were selected and their expression levels were compared using RPKM values.

We focused on contig 49487, which is most specifically expressed in dillapiole-containing perilla (Table 1). PCR-amplification was performed for the target sequence, using cDNA as a template. cDNA was obtained by reverse transcription of RNA isolated from fresh young perilla leaves (strain 5316) using a RNeasy Plant Mini Kit (Qiagen), and reverse transcription was performed using RevTra Ace (Toyobo) with primer add2 (5′-CCACGCGTCGACTACTTTTTTTTTTTTTTT-3′). Primers for PCR-amplification were designed based on sequence information from the EST database. The forward primer 49487-f1 (5′-TCCGTTCCGTTCCTTCAGAGATCTCGCG-3′) and reverse primer 49487-r1 (5′-GGATGCCTTATCAGTTCAGTCATTGCC-3′) were used for amplifying contig 49487. The reaction mixture contained 0.2 µM primer, 0.2 mM dNTPs, and 0.025 U µL^−1^ Blend Taq (Toyobo). The temperature program started at 94 °C for 30 s, followed by 30 cycles of 51 °C for 30 s, 72 °C for 60 s, and a final elongation at 72 °C for 60 s. The size of contig 49487 corresponded to the sequence information from the EST database and RACE methods were used to obtain its full-length sequence. 3′-RACE was performed with the primers add2 and 49487-f2 (5′-AGAAGGTCGGCACAATGGTCAGCTCCATC-3′), and then nested PCR was performed with the primers amm (5′-GGCCACGCGTCGACTAC-3′) and 49487-f3 (5′-TGTAGGTCTGCGTTCGGCACGGTGTGCAAG-3′) in the same reaction mixture as described above, with a temperature program starting at 94 °C for 30 s, followed by 30 cycles of 55 °C (52 °C for nested PCR) for 30 s, 72 °C for 60 s, and a final elongation at 72 °C for 60 s. The reaction products were electrophoresed in agarose gel, purified using NucleoSpin Gel and PCR clean-up (Macherey-Nagel), and ligated to the vector pTA2 (Toyobo). Sequences were confirmed using FASMAC. For 5′-RACE, reverse transcription was performed as described above with primer 49487-r2 (5′-GTAATTGTACCAGCAGACCTCTAGG-3′) or primer 49487-r3 (5′-GGATCATTGTCTTTGAGGACTTCCTTC-3′) and the reaction products were purified using NucleoSpin Gel and PCR clean-up. After the addition of poly C using 0.6 U µL^−1^ TdT (Invitrogen) and 0.2 mM dCTP (Toyobo), PCR was performed with the primers 5ann (5′-GGCCACGCGTCGACTAGTACGGG(I)(I)GGG(I)(I)GGG(I)(I)G-3′) and 49487-r3 or 49487-r4 (5′-CGATGACACGAGGAACGAGTCGAC-3′), and then nested PCR was performed with the primers amm and 49487-r3 in the same reaction mixture as described above, with a temperature program starting at 94 °C for 100 s, followed by 25 cycles (30 cycles for nested PCR) of 94 °C for 30 s, 50 °C (50.5 °C for nested PCR) for 30 s, 72 °C for 60 s, and a final elongation at 72 °C for 60 s.

### 4.3. Heterologous Expression of Pf-49487 in Saccharomyces cerevisiae

*Pf-49487* was ligated into yeast expression vector pGYR-SpeI containing the *S. cerevisiae* NADPH-P450 reductase gene and a SpeI-cloning site [17]. The full-length sequence of *Pf-49487* was amplified by PCR with the forward primer 49487-f4 (5′-ACTAGTATGGAGTCCGATCTCGCAACTG-3′) and the reverse primer 49487-r5 (5′-ACTAGTTCACGGTGATGTCGGTTCAAATGG-3′) in a reaction mixture containing 0.3 µM primer, 0.2 mM dNTPs, and 0.02 U µL^−1^ KOD-Plus (Toyobo) using a temperature program starting at 94 °C for 100 s, followed by 25 cycles of 94 °C for 15 s, 52.7 °C for 30 s, 68 °C for 90 s, and final elongation at 68 °C for 90 s. The amplified sequence was ligated into pTA2 and the nucleotide sequence was confirmed, then *Pf-49487* was ligated into pGYR-SpeI and the vector was introduced into *S. cerevisiae* strain AH22 using the LiCl method. The recombinant yeast cells were cultivated, and the microsomal fraction was prepared as previously described [23]. The microsomal fraction was suspended in 100 mM HEPES/NaOH (pH 7.5) and stored at −80 °C until needed. The reduced CO-difference spectrum was measured with a UV1800 spectrophotometer (Shimadzu) and enzymatic activity was confirmed by the peak at 450 nm [24].

### 4.4. Enzymatic Assays and Gas Chromatography-Mass Spectrometry (GC-MS) Analyses

Enzymatic reactions were performed in 1-mL volumes in screw-capped glass tubes. Each reaction mixture was composed of 50 mM HEPES/NaOH (pH 7.5), 0.5 mM NADPH, 0.2 mM substrate, and 1.98 mg mL^−1^ enzyme preparation. After incubation at 30 °C for 15 min, 12 nmol of eugenol was added as an internal standard, and the reaction mixture was extracted three times with 2 mL of pentane. The pentane fractions were combined, dehydrated with MgSO_4_, concentrated under nitrogen, and analyzed by gas chromatography-mass spectrometry (GC-MS) instrument (6850GC/5975MSD, Agilent Technologies or GCMS-QP2020 NX, Shimadzu). The compounds were separated on a DB-WAX column (60 m × 0.25 mm × 0.25 μm, Agilent Technologies, Santa Clara, CA, USA) under the following conditions: injector, 180 °C; oven program starting at 100 °C, increasing at 5 °C min^−1^ to 220 °C, and holding at this temperature for 60 min. Helium was used as the carrier gas and column flow was 1.0 mL min^−1^. The compounds were identified by comparing their retention times and mass spectra with authentic standards, synthesized samples, or a MS data library (NIST11 or NIST17; National Institute of Standards and Technology).

For kinetic analyses, 0.78 mg mL^−1^ enzyme preparations were incubated with myristicin concentrations ranging from 0.2 to 10 µM. The reaction mixtures were treated as described above and analyzed by GC-FID (G5000, Hitachi, Tokyo, Japan). Compounds were separated on an InertCap WAX column (60 m × 0.25 mm × 0.25 μm, GL Sciences) or a DB-WAX column (60 m × 0.25 mm × 0.25 μm, Agilent Technologies) under the following conditions: injector, 180 °C; FID, 220 °C; oven program starting at 100 °C, increasing at 5 °C min^−1^ to 220 °C, and holding at this temperature for 40 min. Assays were repeated independently three times. The P450 concentration in the enzyme preparation was determined from the reduced CO-difference spectrum using a differential absorption coefficient of 91 mM^−1^ cm^−1^ [24].

The optimum pH (ranging from 6.0 to 9.0) was analyzed using 0.2 mM myristicin and 1.65 mg mL^−1^ enzyme preparation. Assays were performed in the same manner as the kinetic analyses.

### 4.5. Quantitative RT-PCR

The differences in the expression levels of *Pf-49487* between different perilla oil types were determined using quantitative RT-PCR. Total RNA was isolated from fresh young perilla leaves (strains 12, 25, and 5316) using the method described above. First-strand cDNAs were synthesized from 1 µg total RNA using ReverTra Ace and oligo(dT) primer (Takara), and were then purified using NucleoSpin Gel and PCR clean-up. Quantitative RT-PCR was performed with StepOnePlus (Applied Biosystems) using THUNDERBIRD SYBR qPCR Mix (Toyobo) following the manufacturers’ protocols, with a temperature program starting at 95 °C for 60 s, followed by 40 cycles of 95 °C for 15 s, 60 °C for 60 s. The forward primer 49487-f (5′-TGGTGCCTCTCATAATGCTG-3′) and the reverse primer 49487-r (5′-TCTGAAGGAACGGAACGGAG-3′) were used to amplify *Pf-49487*, and the forward primer Histone-f (5′-TCACGAACAAGCCTTTGGAA-3′) and the reverse primer Histone-r (5′-AAGCCTCACCGTTACCGTC-3′) were used to provide histone transcripts as the internal control for PCR. Quantitation was performed using the 2^−ΔΔCT^ method. Total RNA was extracted in duplicate and reactions were performed independently three times.

### 4.6. GC-MS Analysis of Perilla Oil

Perilla essential oils were obtained by extracting approximately 300 g of fresh perilla leaves (strains 10, 12, and 5316) with diethyl ether overnight at 4 °C. The oils were concentrated, dehydrated, and analyzed by GC-MS as described above. Because these analyses were performed to confirm the presence of monohydrate intermediates, the oils were highly concentrated; consequently, we focused on the retention times of the synthesized compounds (Table 2). The chemical profiles of each perilla strain were previously reported [4,16].

### 4.7. Chemicals

Chemical reagents were purchased from Nacalai Tesque or Fujifilm Wako Pure Chemical. Compounds 5, 8, 14, 20, 23, and 30 were synthesized as described below (Figure 8). NMR, MS, and IR data are shown in Appendix A. Compound 31 was synthesized as previously reported [25].

All melting points were uncorrected. Silica gel was used for column chromatography. Commercially available solvents and reagents were purchased and used without purification.

#### 4.7.1. Synthesis of 6-Allyl-2,3-dimethoxyphenol (5)

2-Benzenesulfonyl-1,3-dibromopropane [219500-61-5] (1): The title compound was prepared according to the reported procedure as colorless needles [26,27] of mp 93–94 °C (EtOAc/hexane).1-(2-Benzenesulfonylallyloxy)-2,3-dimethoxybenzene (2): 2,3-Dimethoxyphenol (262 mg, 1.70 mmol) was dissolved in dry acetone (17 mL), and to the solution was added K_2_CO_3_ (705 mg, 5.10 mmol). The resulting suspension was stirred at room temperature for 10 min, and 2-benzenesulfonyl-1,3-dibromopropane (1) (640 mg, 1.87 mmol) and KI (7 mg, 0.04 mmol) were added to the suspension. After 3 h, the mixture was filtered through Celite, and the residue was successively washed with EtOAc. The combined filtrate and washings were concentrated in vacuo and purified by column chromatography (hexane/EtOAc 4:1) to give 2,3-dimethoxyphenol (45 mg, 17%) and the title compound (453 mg, 80%) as pale-yellow oils.6-(2-Benzenesulfonylallyl)-2,3-dimethoxyphenol (3): 1-(2-Benzenesulfonylallyloxy)-2,3-dimethoxybenzene (2) (436 mg, 1.30 mmol) was dissolved in toluene (1 mL) and heated at 165 °C for 12 h under microwave irradiation. The mixture was concentrated in vacuo and purified by column chromatography (hexane/EtOAc 7:3) to give the title compound (398 mg, 92%) as a pale-yellow solid: mp 108–109 °C (EtOAc/hexane).3-Benzenesulfonyl-7,8-dimethoxychromane (4): To a solution of 6-(2-benzenesulfonylallyl)-2,3-dimethoxyphenol (3) (144 mg, 0.430 mmol) in dry acetone (4 mL), was added K_2_CO_3_ (89 mg, 0.64 mmol), and the mixture was heated under reflux for 5 h. The mixture was diluted with EtOAc and washed with H_2_O. The aqueous layer was extracted with EtOAc, and the combined organic layers were washed with brine, dried over Na_2_SO_4_, and concentrated in vacuo to give the title compound as a white solid (140 mg). The crude product was used in the next step without further purification. The title compound was characterized after recrystallization from EtOAc/hexane, which gave colorless needles of mp 132–134 °C.6-Allyl-2,3-dimethoxyphenol [450357-58-1] (5): To a solution of the crude 3-benzenesulfonyl-7,8-dimethoxychromane (4) (140 mg) in EtOAc/MeOH (2:1, 4.5 mL), was added 5% sodium amalgam (0.93 g, 2.0 mmol) at room temperature, and the mixture was stirred for 4 h. The reaction was quenched by the addition of solid citric acid (0.39 g), and the mixture was diluted with EtOAc, washed with saturated aqueous NaHCO_3_, dried over Na_2_SO_4_, and concentrated in vacuo. The residue was purified by column chromatography (hexane/EtOAc 9:1) to give the title compound (65 mg, 78% over 2 steps) as a pale-yellow oil.

#### 4.7.2. Alternative Synthesis of 6-Allyl-2,3-dimethoxyphenol (5)

1-(Allyloxy)-2,3-dimethoxybenzene [380621-78-3] (6): The title compound was prepared from 2,3-dimethoxyphenol (982 mg, 6.37 mmol) according to the reported procedure [28]. After purification by column chromatography (hexane/EtOAc 9:1), the title compound (1.11 g, 90%) was obtained as a colorless oil.6-Allyl-2,3-dimethoxyphenol [450357-58-1] (5) and 4-Allyl-2,3-dimethoxyphenol [29445-64-5] (7): A solution of 1-allyloxy-2,3-dimethoxtbenzene (6) (528 mg, 2.72 mmol) in decaline (0.5 mL) was heated at 200 °C under microwave irradiation for 10 h. After cooled to ambient temperature, the solution was directly purified by column chromatography (hexane/EtOAc 9:1) to give 6-allyl-2,3-dimethoxyphenol (103 mg, 20%) as a pale-yellow oil and a 63:37 mixture of 6-allyl-(5) and 4-allyl-2,3-dimethoxyphenol (7) (367 mg, 44% and 26%, respectively) as a pale-yellow oil. 4-Allyl-2,3-dimethoxyphenol (7) was characterized as a mixture with the other isomer.

#### 4.7.3. Synthesis of 2-Allyl-4,5-dimethoxyphenol (8)

2-Allyl-4,5-dimethoxyphenol [59893-87-7] (8) and 2-Allyl-3,4-dimethoxyphenol [66967-26-8] (9): 4-allyloxy-1,2-dimethoxybenzene [29] (515 mg, 2.65 mmol) was dissolved in decaline (0.5 mL) and heated at 200 °C under microwave irradiation for 12 h. After cooled to ambient temperature, the solution was directly purified by column chromatography (hexane/EtOAc 3:1) to give 2-allyl-4,5-dimethoxyphenol (8) (68 mg, 13%) as a pale-yellow solid along with a 91:9 mixture of the title compounds (411 mg, 73% and 7%, respectively) as a pale-yellow oil. 2-Allyl-3,4-dimethoxyphenol (9) was partially separated as a pale-yellow oil from the mixture by another column chromatography (hexane/EtOAc 3:1) for characterization. 2-Allyl-4,5-dimethoxyphenol: mp 35–36 °C (Et_2_O/hexane) (lit. 42–42.5 °C) [30].

#### 4.7.4. Synthesis of 5-Allyl-2,3-dimethoxyphenol (14)

3-*tert*-Butyldimethylsiloxy-4,5-dimethoxybenzaldehyde [122271-47-0] (10): To a solution of 3-hydroxy-4,5-dimethoxybenzaldehyde (500 mg, 2.74 mmol) in dry N,N-dimethylformamide (DMF) (3 mL), were added *tert*-butylchlorodimethylsilane (496 mg, 3.29 mmol) and imidazole (466 mg, 6.84 mmol). The mixture was stirred at room temperature for 16 h, diluted with EtOAc, washed with saturated aqueous NaHCO_3_ and brine, dried over Na_2_SO_4_, and concentrated in vacuo. The resulting pale-yellow oil, containing mainly the title compound, was used for the next step without further purification. The ^1^H NMR was consistent with that reported [31].3-*tert*-Butyldimethylsiloxy-4,5-dimethoxybenzenemethanol [111394-55-9] (11): To a solution of the crude 3-*tert*-butyldimethylsiloxy-4,5-dimethoxybenzaldehyde (10) in EtOH (24 mL) cooled in an ice–water bath, was portion-wise added NaBH_4_ (104 mg, 2.75 mmol). After 15 min, water was added to the mixture, and most of EtOH was removed from the mixture by evaporation. The mixture was extracted three times with EtOAc, and the combined organic layers were washed with brine, dried over Na_2_SO_4_, and concentrated in vacuo. The resulting residue was purified by column chromatography (hexane/EtOAc 3:1) to give the title compound (563 mg, 69% over 2 steps) as a white solid. Recrystallization from hexane gave colorless needles of mp 57–58 °C. The ^13^C NMR was identical to that reported, while all the ^1^H NMR chemical shifts differed by 0.18 ppm from the reported values [32].3-*tert*-Butyldimethylsiloxy-4,5-dimethoxybenzyl Bromide [111394-56-0] (12): To a solution of 3-*tert*-butyldimethylsiloxy-4,5-dimethoxybenzenemethanol (11) (315 mg, 1.06 mmol) in dry CH_2_Cl_2_ (10 mL) cooled in an ice–water bath, were added CBr_4_ (420 mg, 1.27 mmol) and Ph_3_P (333 mg, 1.27 mmol). The cooling bath was removed, and the mixture was stirred at room temperature for 20 min. Volatile materials were removed from the mixture by evaporation, and the residue was purified by column chromatography (hexane to hexane/EtOAc 97:3) to give the title compound (305 mg, 80%) as a pale-yellow oil.5-Allyl-1-*tert*-butyldimethylsiloxy-2,3-dimethoxybenzene (13): To a mixture of 3-*tert*-butyldimethylsiloxy-4,5-dimethoxybenzyl bromide (12) (396 mg, 1.10 mmol), CuI (21 mg, 0.11 mmol), and 2,2’-bipyridine (17 mg, 0.11 mmol) in Et_2_O (1 mL) cooled in an ice–water bath, was dropwise added a 1.0 M tetrahydrofuran (THF) solution of vinylmagnesium bromide (1.6 mL, 1.6 mmol). The cooling bath was removed, and the mixture was stirred at room temperature for 1 h. The reaction was quenched by the addition of saturated aqueous NH_4_Cl and then 28% aqueous NH_3_. After stirred for 30 min, the whole was extracted three times with Et_2_O. The combined organic layers were washed with brine, dried over Na_2_SO_4_, and concentrated in vacuo. The residue was purified by column chromatography (hexane/EtOAc 49:1) to give the title compound (266 mg, 78%) as a pale-yellow oil.5-Allyl-2,3-dimethoxyphenol [76773-99-4] (14): To a stirred solution of 5-allyl-1-*tert*-butyldimethylsiloxy-2,3-dimethoxybenzene (13) (74 mg, 0.24 mmol) in THF (1 mL), was added a 1.0 M THF solution of tetrabutylammonium fluoride (TBAF) (0.24 mL, 0.24 mmol). After 30 min, 0.5 M aqueous citric acid (1 mL) was added to the mixture, and the whole was extracted three times with EtOAc. The combined organic layers were washed with saturated aqueous NaHCO_3_ and brine, dried over Na_2_SO_4_, and concentrated in vacuo. The residue was purified by column chromatography (hexane/EtOAc 9:1) to give the title compound (43 mg, 91%) as a pale-yellow solid of mp 38–40 °C.

#### 4.7.5. Synthesis of 6-Allyl-4-methoxy-1,3-benzodioxol-5-ol (20)

4-Methoxy-1,3-benzodioxole [1817-95-4] (15): The title compound was prepared from 3-methoxycatechol according to the reported procedure [33] in 85% yield as colorless blocks of mp 41–42 °C.4-Methoxy-1,3-benzodioxole-5-carbaldehyde [5779-99-7] (16) and 7-Methoxy-1,3-benzodioxole-4-carbaldehyde [23731-55-7] (17): Dry DMF (0.64 mL, 8.3 mmol) and freshly distilled POCl_3_ (0.77 mL, 8.3 mmol) were mixed at 100 °C for 2 h. To the mixture, 4-methoxy-1,3-benzodioxole (15) (500 mg, 3.29 mmol) was added, and the mixture was heated at 100 °C. After 2 h, the reaction was quenched by the addition of ice, and the whole was extracted three times with Et_2_O. The combined organic layers were washed with water, saturated aqueous NaHCO_3_, and brine, dried over Na_2_SO_4_, and concentrated in vacuo to give a crude mixture of the title compounds (3:2) as a pale brown solid. The two isomers were separated by column chromatography (hexane/Et_2_O 4:1, *R_f_* = 0.24 and 0.12, respectively). 4-Methoxy-1,3-benzodioxole-5-carbaldehyde (16): 44% yield. Colorless needles of mp 107–108 °C (EtOH). 7-Methoxy-1,3-benzodioxole-4-carbaldehyde (17): 29% yield. Colorless needles of mp 85–86 °C (EtOH).4-Methoxy-1,3-benzodioxol-5-ol [23504-78-1] (18): To a solution of 4-methoxy-1,3-benzodioxole-5-carbaldehyde (16) (100 mg, 0.555 mmol) in MeOH (1 mL), were added conc. H_2_SO_4_ (0.01 mL, 0.2 mmol) and 30% H_2_O_2_ (0.085 mL, 0.83 mmol). The mixture was stirred at room temperature for 2.5 h, and saturated aqueous NaHCO_3_ was added to the mixture. The whole was extracted three times with EtOAc, and the combined organic layers were washed with brine, dried over Na_2_SO_4_, and concentrated in vacuo. The residual solid was purified by column chromatography (hexane/EtOAc 9:1) to give the title compound (72 mg, 78%) as a white solid: *R_f_* = 0.29 (hexane/EtOAc 4:1). Colorless plates of mp 58–59 °C (Et_2_O/hexane).5-Allyloxy-4-methoxy-1,3-benzodioxole [23731-59-1] (19): A mixture of 4-methoxy-1,3-benzodioxol-5-ol (18) (67 mg, 0.40 mmol), K_2_CO_3_ (111 mg, 0.800 mmol), and allyl bromide (0.05 mL, 0.6 mmol) in dry acetone (4 mL) was heated under reflux for 4.5 h, and water was added to the mixture. The whole was extracted three times with EtOAc, and the combined organic layers were washed with 15% aqueous NaOH twice and brine, dried over Na_2_SO_4_, and concentrated in vacuo. The residue was purified by column chromatography (hexane/EtOAc 95:5) to give the title compound (71.6 mg, 86%) as a pale-yellow oil: *R_f_* = 0.25 (hexane/EtOAc 19:1).6-Allyl-4-methoxy-1,3-benzodioxol-5-ol [23731-60-4] (20): A solution of 5-allyloxy-4-methoxy-1,3-benzodioxole (19) (58 mg, 0.28 mmol) in a 1:2 mixture of N-methyl-2-pyrrolidone (NMP) and decaline (1.5 mL) was heated at 200 °C under microwave irradiation for 8 h. After cooled to ambient temperature, the mixture was diluted with EtOAc, washed three times with water and with brine, dried over Na_2_SO_4_, and concentrated in vacuo. The residue was purified by column chromatography (hexane/EtOAc 9:1) to give the title compound (46 mg, 78%) as a pale-yellow oil: *R_f_* = 0.44 (hexane/EtOAc 4:1).

#### 4.7.6. Synthesis of 5-Allyl-7-methoxy-1,3-benzodioxol-4-ol (23)

7-Methoxy-1,3-benzodioxol-4-ol [23812-54-6] (21): To a solution of 7-methoxy-1,3-benzodioxole-4-carbaldehyde (17) (100 mg, 0.555 mmol) in MeOH (1 mL), conc. H_2_SO_4_ (0.01 mL, 0.2 mmol) and 30% H_2_O_2_ (0.085 mL, 0.83 mmol) were added. The mixture was stirred at room temperature for 6 h, and saturated aqueous NaHCO_3_ was added to the mixture. The whole was extracted three times with EtOAc, and the combined organic layers were washed with brine, dried over Na_2_SO_4_, and concentrated in vacuo. The residual solid was purified by column chromatography (hexane/EtOAc 3:1) to give the title compound (85 mg, 92%) as a white solid: *R_f_* = 0.13 (hexane/EtOAc 4:1). Colorless needles of mp 104–105 °C (CHCl_3_/hexane).4-Allyloxy-7-methoxy-1,3-benzodioxole [23731-70-6] (22): A mixture of 7-methoxy-1,3-benzodioxol-4-ol (21) (82 mg, 0.49 mmol), K_2_CO_3_ (135 mg, 0.977 mmol), and allyl bromide (63 μL, 0.73 mmol) in dry acetone (2.5 mL) was heated under reflux for 3 h, and water was added to the mixture. The whole was extracted three times with EtOAc, and the combined organic layers were washed with 15% aqueous NaOH twice and brine, dried over Na_2_SO_4_, and concentrated in vacuo. The residue was purified by column chromatography (hexane/EtOAc 95:5) to give the title compound (83.7 mg, 82%) as a pale-yellow oil: *R_f_* = 0.20 (hexane/EtOAc 19:1).5-Allyl-7-methoxy-1,3-benzodioxol-4-ol [76773-99-4] (23): A solution of 4-allyloxy-7-methoxy-1,3-benzodioxole (22) (82 mg, 0.39 mmol) in a 1:4 mixture of NMP and decaline (1 mL) was heated at 200 °C under microwave irradiation for 8 h. After cooled to ambient temperature, the mixture was diluted with EtOAc, washed three times with water and with brine, dried over Na_2_SO_4_, and concentrated in vacuo. The residue was purified by column chromatography (hexane/EtOAc 4:1) to give the title compound (68 mg, 84%) as a pale-yellow solid: *R_f_* = 0.24 (hexane/EtOAc 4:1). Colorless needles of mp 84–85 °C (EtOAc/hexane).

#### 4.7.7. Synthesis of 5-Allyl-6,7-dimethoxy-1,3-benzodioxol-4-ol (30)

6,7-Dimethoxy-1,3-benzodioxol-4-ol [22934-71-0] (24): A mixture of 6,7-dimethoxy-1,3-benzodioxole-4-carbaldehyde (368 mg, 1.75 mmol), 30% H_2_O_2_ (0.27 mL, 2.6 mmol), and conc. H_2_SO_4_ (0.12 mL, 2.3 mmol) in MeOH (3.5 mL) was stirred at room temperature for 3 h. After addition of saturated aqueous NaHCO_3_, the whole was extracted three times with EtOAc, and the combined organic layers were washed with brine, dried over Na_2_SO_4_, and concentrated in vacuo. The residual pale-yellow solid (1.38 g, 78%) was used as a crude title compound without further purification to the next step.: *R_f_* = 0.22 (hexane/EtOAc 2:1). Colorless needles of mp 111–112 °C (EtOAc/hexane).4-Hydroxy-6,7-dimethoxy-1,3-benzodioxol-5-carbadehyde (25): A mixture of 6,7-dimethoxy-1,3-benzodioxol-4-ol (24) (589 mg, 2.97 mmol) and hexamethylenetetramine (834 mg, 5.94 mmol) in refluxing trifluoroacetic acid (TFA) (3 mL) was stirred for 24 h [34]. H_2_O was added, and the solution was stirred for further 30 min at 60 °C. After being cooled to room temperature, the whole was extracted three times with EtOAc, and the combined organic layers were washed with brine, dried over Na_2_SO_4_, and concentrated in vacuo. The residual solids were purified by column chromatography (hexane/EtOAc 7:1) to give the title compound (546 mg, 81%) as a pale-yellow solid: *R_f_* = 0.25 (hexane/EtOAc 7:1). Colorless powder of mp 85–87 °C (EtOAc/hexane).4-*tert*-Butyldimethylsilyloxy-6,7-dimethoxy-1,3-benzodioxole-5-carbadehyde (26): To a mixture of 4-hydroxy-6,7-dimethoxy-1,3-benzodioxole-5-carbaldehyde (25) (1.20 g, 5.31 mmol), 4-dimethylaminopyridine (DMAP) (130 mg, 1.06 mmol), and Et_3_N (1.5 mL, 11 mmol) in CH_2_Cl_2_ (27 mL), was added *tert*-butylchlorodimethylsilane (1.20 g, 7.96 mmol). The mixture was stirred at room temperature for 3 h. After addition of saturated aqueous NaHCO_3_, the whole was extracted three times with EtOAc, and the combined organic layers were washed with brine, dried over Na_2_SO_4_, and concentrated in vacuo. The residual pale-yellow oil (2.14 g) was used as a crude title compound without further purification to the next step.4-*tert*-Butyldimethylsilyloxy-6,7-dimethoxy-1,3-benzodioxole-5-methanol (27): To a solution of the crude 4-*tert*-butyldimethylsiloxy-6,7-dimethoxy-1,3-benzodioxole-5-carbaldehyde (26) (2.14 g) in EtOH (30 mL) was added NaBH_4_ (714 mg, 18.8 mmol). The mixture was stirred at room temperature for 30 min. After addition of water, the whole was extracted three times with EtOAc, and the combined organic layers were washed with brine, dried over Na_2_SO_4_, and concentrated in vacuo. The residual solids were purified by column chromatography (hexane/EtOAc 9:1 to 5:1) to give the title compound (1.73 g, 95% in two steps) as a colorless oil: *R_f_* = 0.23 (hexane/EtOAc 7:1).5-Bromomethyl-4-*tert*-butyldimethylsilyloxy-6,7-dimethoxy-1,3-benzodioxole (28): To a mixture of 4-*tert*-butyldimethylsiloxy-6,7-dimethoxy-1,3-benzodioxole-5-methanol (27) (450 mg, 1.31 mmol) in CH_2_Cl_2_ (13 mL) was added PBr_3_ (0.14 mL, 1.4 mmol). The mixture was stirred at room temperature for 30 min. After addition of water, the whole was extracted three times with CHCl_3_, and the combined organic layers were washed with brine, dried over Na_2_SO_4_, and concentrated in vacuo. The residual pale-yellow oil (552 mg) was used as a crude title compound without further purification to the next step.5-Allyl-4-*tert*-butyldimethylsilyloxy-6,7-dimethoxy-1,3-benzodioxole (29): To a mixture of the crude 5-bromomethyl-4-*tert*-butyldimethylsiloxy-6,7-dimethoxy-1,3-benzodioxole (28) (552 mg, 1.56 mmol), CuI (50 mg, 0.26 mmol), and 2,2’-bipyridine (41 mg, 0.26 mmol) in Et_2_O (4 mL) was added a 1.0 M THF solution of vinylmagnesium bromide (2.0 mL, 2.0 mmol). The mixture was stirred at room temperature for 30 min. After addition of water, the whole was extracted three times with EtOAc, and the combined organic layers were washed with brine, dried over Na_2_SO_4_, and concentrated in vacuo. The residual solids were purified by column chromatography (hexane/EtOAc 50:1) to give the title compound (313 mg, 68% over 2 steps) as a pale-yellow oil: *R_f_* = 0.52 (hexane/EtOAc 7:1).5-Allyl-6,7-dimethoxy-1,3-benzodioxol-4-ol (30): To a solution of 5-allyl-4-*tert*-butyldimethylsiloxy-6,7-dimethoxy-1,3-benzodioxole (29) (690 mg, 1.96 mmol) in THF (20 mL) was added a 1.0 M THF solution of TBAF (1.96 mL, 2.0 mmol). The mixture was stirred at room temperature for 30 min. After addition of water, the whole was extracted three times with EtOAc, and the combined organic layers were washed with brine, dried over Na_2_SO_4_, and concentrated in vacuo. The residual solids were purified by column chromatography (hexane/EtOAc 5:1 to 4:1) to give the title compound (462 mg, 99%) as a pale-yellow solid: *R_f_* = 0.28 (hexane/EtOAc 4:1). Colorless needles of mp 65–67 °C (EtOAc/hexane).

## 5. Conclusions

An enzyme catalyzing a reaction in a proposed synthetic pathway of PP-type perilla oil components, namely, the conversion of myristicin into a monohydrate intermediate of dillapiole (compound 20) by regiospecific hydroxylation, was cloned. The potentially harmful to humans (*E*)-Asarone is structurally similar to PP-type oil components and has been identified in perilla plants grown in China [12,13]. The methoxy group of (*E*)-asarone may result from the hydroxylation of an intermediate. The position of the double bond in the side chain of (*E*)-asarone differs from that of PP-type oil components and thus *Pf-49487* is unlikely to be involved in the biosynthesis of (*E*)-asarone. However, our findings regarding *Pf-49487* may help elucidate the (*E*)-asarone biosynthetic pathway and aid in the quality assessment and safety evaluation of perilla products.

Little is known about the biosynthetic pathways of phenylpropanoid volatile components. Only a few enzymes involved in the synthesis of compounds such as anethole and eugenol have been reported, and enzymes involved in the synthesis of compounds such as apiole, myristicin, and asarone have not been characterized. These components are important plant flavor compounds and are also used as medicines and food supplements due to their pharmacological actions. Some of these compounds are genotoxic or carcinogenic [8]. Further elucidation of the biosynthetic mechanisms of phenylpropanoid volatile components may contribute to the genetic manipulation and enzymatic production of useful compounds.

## Figures and Tables

**Figure 1 plants-09-00577-f001:**
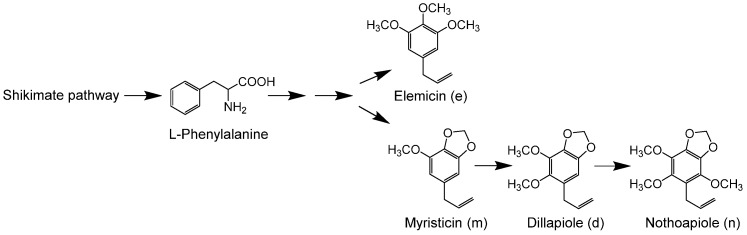
Putative biosynthetic pathway of phenylpropene (PP)-type oils in perilla. In the text, the oil types of each strain of perilla are represented as PP-em, PP-m, PP-md, PP-emd, and PP-mdn, meaning that each strain mainly contains elemicin + myristicin, myristicin, myristicin + dillapiole, elemicin + myristicin + dillapiole, and myristicin + dillapiole + nothoapiole, respectively.

**Figure 2 plants-09-00577-f002:**
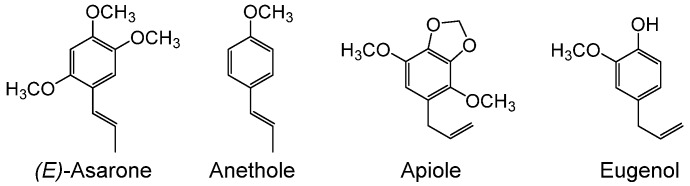
Chemical structures of several phenylpropanoid volatile compounds.

**Figure 3 plants-09-00577-f003:**
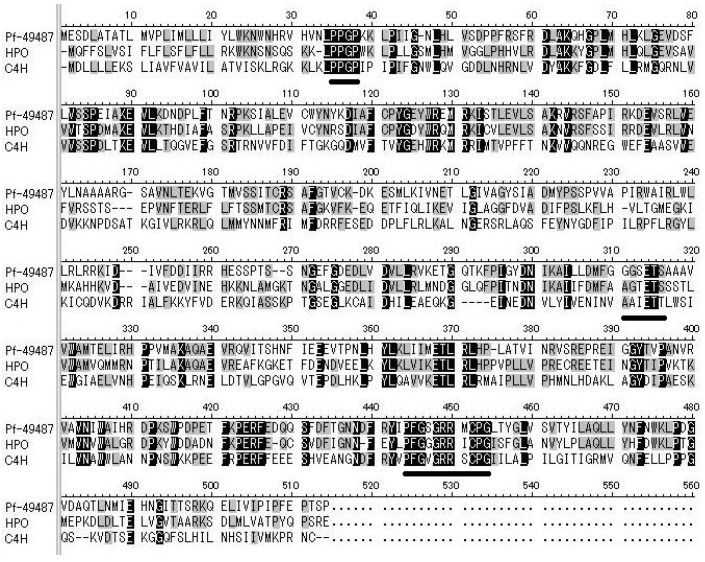
Alignment of the amino acid sequences of *Pf-49487*. HPO: *Hyoscyamus muticus* premnaspirodiene oxygenase (GenBank Accession No. EF569601), C4H: cinnamate 4-hydroxylase *(Arabidopsis thaliana)* (GenBank Accession No. NP180607). The following conserved regions of P450 are underlined: the proline-rich region, the oxygen-binding pocket, and the heme-binding region from upstream of the sequence. Black background indicates 100% amino acid identity among the three clones and gray background indicates greater than 50% amino acid identity.

**Figure 4 plants-09-00577-f004:**
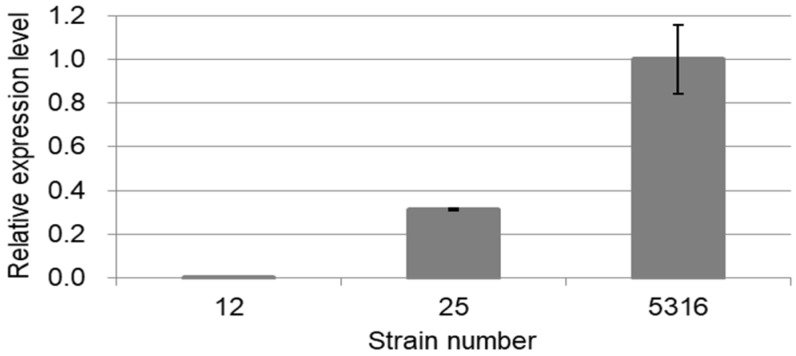
The relative expression levels of *Pf-49487* in strains 12, 25, and 5316. Error bars indicate SE of triplicate analyses of two independent samples.

**Figure 5 plants-09-00577-f005:**
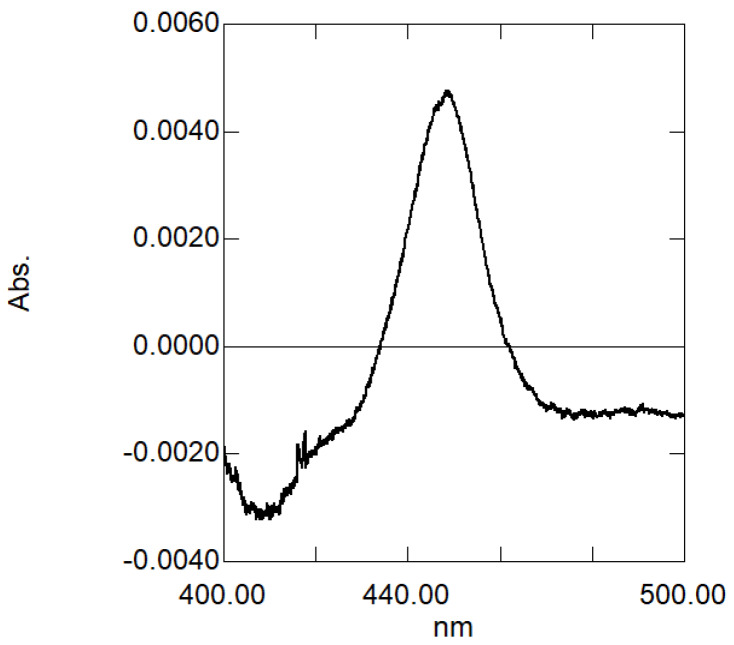
The reduced carbon monoxide (CO)-difference spectrum of the microsomal fraction of yeast harboring the *Pf-49487* expression plasmid.

**Figure 6 plants-09-00577-f006:**
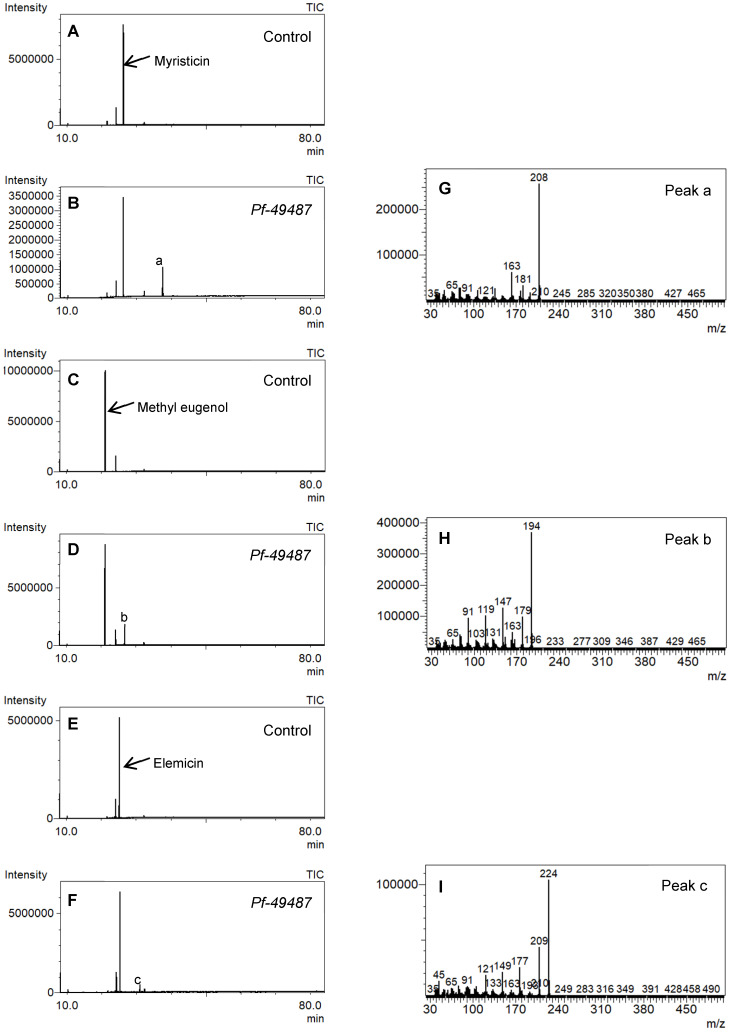
Gas chromatography-mass spectrometry (GC-MS) profiles of the reaction products. (**A**–**F**), Total ion chromatograms of the products obtained using the control (**A**,**C**,**E**) and the microsomal fraction prepared from yeast harboring *Pf-49487* (**B**,**D**,**F**). The substrates are myristicin (**A**,**B**), methyl eugenol (**C**,**D**), and elemicin (**E**,**F**). The controls were incubated without the cofactor NADPH. The peak with a retention time of 24 min is the internal standard, eugenol. (**G**–**I**), Mass spectra of the products formed by *Pf-49487* with myristicin, methyl eugenol, and elemicine as the substrates, respectively.

**Figure 7 plants-09-00577-f007:**
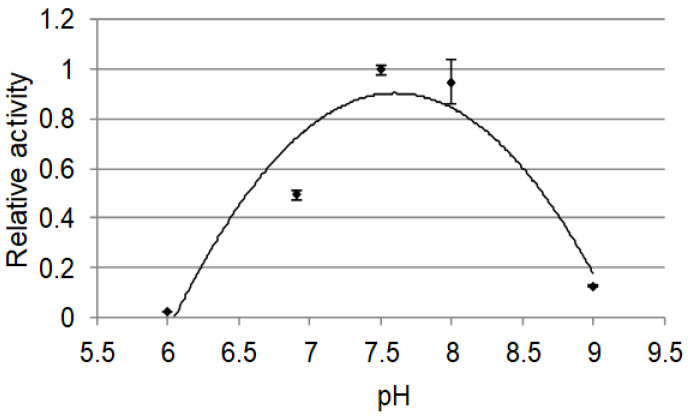
pH-dependent activity of *Pf-49487*. Data are the means ± SE of triplicate analyses.

**Figure 8 plants-09-00577-f008:**
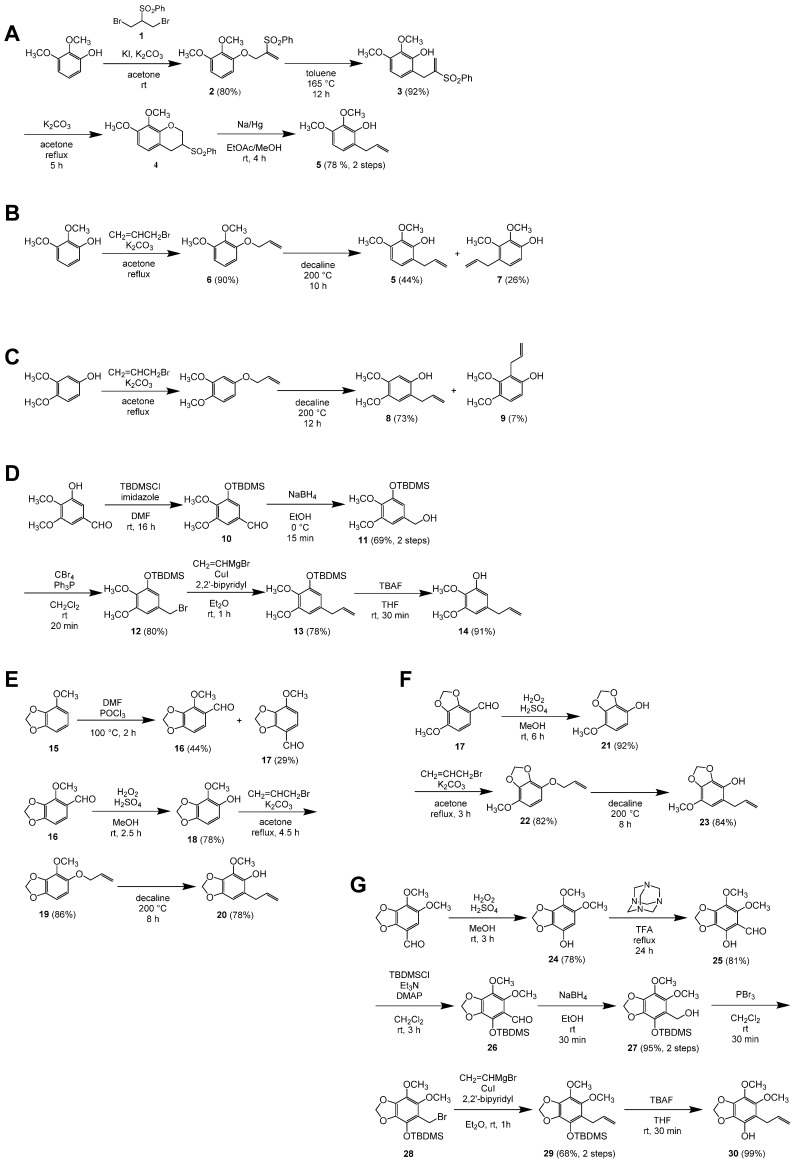
Synthetic scheme of allylphenols. (**A**) Synthesis of 6-allyl-2,3-dimethoxyphenol (5). (**B**) Alternative synthesis of 6-allyl-2,3-dimethoxyphenol (5). (**C**) Synthesis of 2-allyl-4,5-dimethoxyphenol (8). (**D**) Synthesis of 5-allyl-2,3-dimethoxyphenol (14). (**E**) Synthesis of 6-allyl-4-methoxy-1,3-benzodioxol-5-ol (20). (**F**) Synthesis of 5-allyl-7-methoxy-1,3-benzodioxol-4-ol (23). (**G**) Synthesis of 5-allyl-6,7-dimethoxy-1,3-benzodioxol-4-ol (30).

**Table 1 plants-09-00577-t001:** Reads per kilobase of exon per million mapped read (RPKM) values of contig 49487 and 77404. Oil type C means that the strain mainly contains citral, a monoterpene (MT)-type oil component. Contig 49487 had the largest difference in RPKM values between PP-type strains containing dillapiole (i.e., strains 16, 25, and 5316) and others. Contig 77404 showed the second largest difference in RPKM values.

	Strain (Oil Type)					
Contig	10 (PP-em)	12 (PP-m)	16 (PP-md)	25 (PP-emd)	5316 (PP-mdn)	5717 (C)
49487	0.0824	0.0968	68.4	23.7	56.7	0.325
77404	0.260	0.459	1.10	1.40	1.70	0.446

**Table 2 plants-09-00577-t002:** Synthesized allylphenols used in this study.

Formula	Name (Compound Number)	Mass Spectra	Retention Time (min)
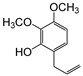	6-Allyl-2,3-dimethoxyphenol (5)	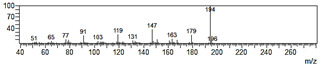	26
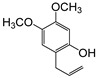	2-Allyl-4,5-dimethoxyphenol (8)	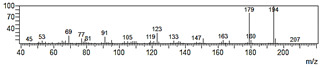	50
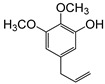	5-Allyl-2,3-dimethoxyphenol (14)	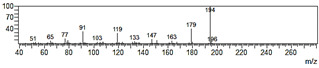	29
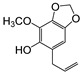	6-Allyl-4-methoxy-1,3-benzodioxol-5-ol (20)	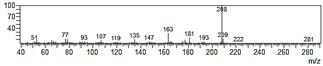	37
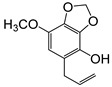	5-allyl-7-methoxy-1,3-benzodioxol-4-ol (23)	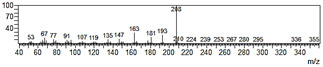	66
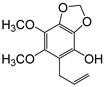	5-Allyl-6,7-dimethoxy-1,3-benzodioxol-4-ol (30)	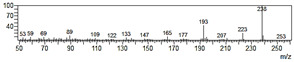	77
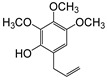	6-Allyl-2,3,4-trimethoxyphenol (31)	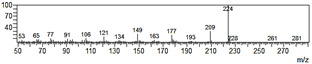	31

**Table 3 plants-09-00577-t003:** Relative activity of *Pf-49487*.

Formula	Substrate	Activity (%)
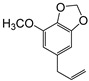	Myristicin	100
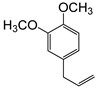	Methyl eugenol	91
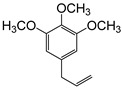	Elemicin	22
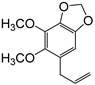	Dillapiole	0
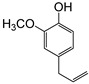	Eugenol	0
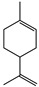	Limonene	0
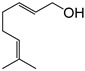	Geraniol	0

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
