# Peer review of "Cloning and Expression of a *Perilla frutescens* Cytochrome P450 Enzyme Catalyzing the Hydroxylation of Phenylpropenes"

_plants, 2020, doi:10.3390/plants9050577_

Round 1
Reviewer 1 Report
In this study, they cloned an enzyme that catalyzes a reaction in a proposed synthetic pathway of
556 PP-type perilla oil components. I think this study may aid the elucidation of generally unexploited biosynthetic pathways of phenylpropanoid volatile components.
I think this manuscript is accept in present form.
Author Response
Thank you for your comments.
Reviewer 2 Report
This is a very interesting article. Well written and scientifically sound.
I have made several comments directly on the pdf that I have attached.
The authors should not have any difficulty addressing those.
The The authors should consider doing a homology model of the P450. I think crystal structure of 5YLW Chain A, Ferruginol synthases could be a good starting point, and then use parts of 6OYU|A Chain A, Cytochrome P450 1B1 to fill in the missing pieces. This model might help explain the regiospecificity of their enzyme.
However, this is not required for acceptance for publication. The data is sufficient for publication as it is.

Author Response
We accepted most of your comments and revised as requested. We thank you for your comments about homology model and we will take it into consideration in our next report.
Reviewer 3 Report
This manuscript describes cloning of the cDNA of the P450 enzyme that hydroxylates phenylpropanoids from perilla and analysis of the substrate specificity of the gene product. This manuscript may contribute in part to the understanding of the biosynthesis of plant-derived volatile phenylpropanoids. The followings will improve this study.
1. The EST data suggest that the cloned P450 is involved in the biosynthesis of volatile phenylpropanoids, the target of our study; however, there is no direct evidence that this is indeed involved in the biosynthesis. Considering the current situation, it seems inadequate to use the present title. I think that authors should rename the title. If authors insist on this title, gene knockout and knockdown experiments are required.
2. I may understand that the identification of the biosynthetic pathway (enzymes) of (E) -asarone and the control of (E) -asarone production is one of motivations of this series of research. However, such a topic is not directly related to the contents of this manuscript. Therefore, the portions related to (E) -asarone seems to be unnecessary in the introduction and In Conclusion and should be moved to Discussion.
3. Lines 143-146: Detection of compounds 14, 20, 30, and 31 in perilla is described in text, but the experimental data or citation for the detection is not described in this manuscript. Moreover, authors describe the formation of only some related compounds in perilla, and do not show whether other compounds are detected or not. This is not preferable because readers read this paper with incomplete information. That part needs improvement.
4. The RPKM values are written in Table 1. I think that such long digits are meaningless for the RPKM values. The authors actually use three digits, 0.968, 23.7, and 56.7, as some values, in text. I recommend authors to unify them into three digits?
5. Lines 270-: In the enzyme assay, the reaction solution without enzyme or NADPH is used as a control when examining the substrate specificity, I think. Please describe it in a correct way.
Author Response
Point 1: The EST data suggest that the cloned P450 is involved in the biosynthesis of volatile phenylpropanoids, the target of our study; however, there is no direct evidence that this is indeed involved in the biosynthesis. Considering the current situation, it seems inadequate to use the present title. I think that authors should rename the title. If authors insist on this title, gene knockout and knockdown experiments are required.
Response 1: The title was renamed.
Point 2: I may understand that the identification of the biosynthetic pathway (enzymes) of (E) -asarone and the control of (E) -asarone production is one of motivations of this series of research. However, such a topic is not directly related to the contents of this manuscript. Therefore, the portions related to (E) -asarone seems to be unnecessary in the introduction and In Conclusion and should be moved to Discussion.
Response 2: The content of (E)-asarone in Perilla Herb is a serious issue for Japanese regulation on medicinal plants so that the biosynthetic pathways relevant to (E)-asarone shall be an interesting topic. Surely, studies on relationships between syntheses of PP-type compounds and (E)-asarone was one of our purposes. Our studies continue to further steps as for this issue in near future. We leave these paragraphs as they are.
Lines 143-146: Detection of compounds 14, 20, 30, and 31 in perilla is described in text, but the experimental data or citation for the detection is not described in this manuscript. Moreover, authors describe the formation of only some related compounds in perilla, and do not show whether other compounds are detected or not. This is not preferable because readers read this paper with incomplete information. That part needs improvement.
Response 3: We described that those compounds were detected and analyzed by GC-MS in lines 311-316.
Point 4: The RPKM values are written in Table 1. I think that such long digits are meaningless for the RPKM values. The authors actually use three digits, 0.968, 23.7, and 56.7, as some values, in text. I recommend authors to unify them into three digits?
Response 4: Table 1 was revised.
Lines 270-: In the enzyme assay, the reaction solution without enzyme or NADPH is used as a control when examining the substrate specificity, I think. Please describe it in a correct way.
Response 5: We described the enzyme assay control in caption of figure 6 (line 176).